# Reliability-Based Design of an Aircraft Wing Using a Fuzzy-Based Metaheuristic

**Seksan Winyangkul** [1], **Suwin Sleesongsom** [2,*] and **Sujin Bureerat** [1]

1 Sustainable and Infrastructure Development Center, Department of Mechanical Engineering, Faculty of Engineering, KhonKaen University, KhonKaen City 40002, Thailand; seksanwin@kkumail.com (S.W.); sujbur@kku.ac.th (S.B.)
2 Department of Aeronautical Engineering, International Academy of Aviation Industry, King Mongkut's Institute of Technology Ladkrabang, Bangkok 10520, Thailand
* Correspondence: suwin.se@kmitl.ac.th; Tel.: +66-02-329-800

**Abstract:** The purpose of this paper is to design aircraft wing using reliability-based design optimization concerned to fuzzy uncertainty variables. A possibilistic safety index-based design optimization (PSIBDO) with fuzzy uncertainties is proposed to overcome difficult tasks from the original probabilistic problem. The design problem is to minimize mass of a composite aircraft wing subject to aeroelastic and structural constraints through consideration of the material properties are the uncertainties. The design variables include aircraft wing structure dimensions. The reliability-based design approach is needed to alleviate such a problem. Due to the complexity of the aircraft wing structures design and aeroelastic analysis, nonprobability-based design is an alternative choice to increase computational efficiency in the design process. The optimum results show the efficiency of our proposed approach.

**Keywords:** reliability-based design; aeroelasticity; optimization; aircraft wing; optimization technique; metaheuristics

## 1. Introduction

Composite materials are increasingly used as aircraft structures, but complexity in design analysis causes uncertainties due to material non-homogenous and thickness variation. It has been shown that a classical deterministic optimization technique is not enough to handle the present requirements due to the presence of the uncertainties that can be found throughout the whole part of the aircraft [1]. The uncertainties can suppress the actual performance of aircraft far from the optimum design and lead to impracticality. The traditional Monte Carlo simulation (MCS) technique has been used to identify the effects of uncertainty on optimum aircraft structure design. However, it utilizes expansive computation for analysis. For this reason, there other techniques are needed to collect data on the uncertainties in aircraft wing structure design. To enhance performance in the aircraft design process, reliability-based design optimization (RBDO) has been proposed.

The reliability method is used to analyze reliability or failure probability in an optimization problem with uncertainties. The failure probability can be obtained by using probabilistic and non-probabilistic techniques. At present, a method in first group is the MCS method, which is used as a base line to quantify uncertainty. Due to the drawbacks mentioned in the previous cause, several techniques in this group are developed, i.e., the first-order and second moment (FOSM), the first-order reliability method (FORM) and the second-order reliability method (SORM) [2]. The various forms are performed to calculate the reliability index of the design space at the most probable point (MPP). The drawback of this technique is that it needs more precise calculation, which makes its inefficient computation. To increase the performance in generating a probability distributed function (PDF) rather than using MCS, the optimum Latin hypercube sampling (OLHS)

with and without infill sampling has been used [3]. The second group, on the other hand, is proposed to solve the reliability-based design without requiring precise distribution of random variables. The most popular approaches are convex set [4], an interval method [5] and fuzzy set theory [6]. The general reliability-based design optimization problem is a double-looped nested problem due to the calculation of probability failure and optimization solving. In cases of a non-probabilistic approach, triple-looped nested problem is needed due to the possibility safety index (PSI) calculation. This technique is called possibility safety index-based design optimization (PSIBDO). The target-performance-based design approach (TPBDA) is proposed to solve the triple-looped problem by performing only the double-loop nested problem [7]. Then the interval perturbation method (IMP) can reduce the double-loop nested problem to a single one by estimating the constraints of the optimization problem [8]. Most techniques are accomplished with a based gradient, which is needed to perform differentiations. Recently, alternative techniques for solving the original triple-loop nest problem can be performed by means multi-objective optimization with evolutionary algorithms (EAs) or metaheuristics (MHs) [1,9,10]. This technique is solved with the single-looped problem. Reliability-based design optimization of aircraft aeroelasticity is a computational burden problem due to complex of aircraft structures and the double-loop nested problem for the probabilistic technique. The problem changes to the triple-looped problem for the non-probabilistic approach. Further, so far, there have been only a few techniques introducing the non-probabilistic reliability index into the aeroelastic optimization design of aircraft wings [11]. A very recent work applied the worse-case scenario to handle uncertainty into classical aircraft wing aeroelastic design [1].

Aeroelastic aircraft wing design results usually deviate by various uncertainties in prototyping, manufacturing, and other stages in the design process. This directly affects aeroelastic characteristics and the safety of flight. Uncertainty in aeroelastic aircraft wing design usually derives from various sources such as, material properties, load, tolerance in manufacturing processes, and flight conditions. The uncertainty sources can be classified in two categories, namely aleatory uncertainty and epistemic uncertainty. The first group occurs due to random physical variation while the second group is caused by lack of knowledge [2]. It is known that aleatory uncertainty can be handled by the probabilistic model, but epistemic uncertainty is preferably quantified by a convex set—a fuzzy set method [9], and anti-optimization [1,10]. It means that the uncertainties of aeroelastic aircraft wing design cannot be quantified by theory alone. Composite aircraft wing structures that have been studied in design optimization are laminate layup in upper and lower skin [12], laminate ply thickness [13] and ribs and spars dimension [14], which resulted in both reduction in structural weight and flutter speed. Probabilistic reliability-based aeroelastic design of wings for aeroelastic tailoring in layup optimization of ply angles uncertain for composite wings [15] when reduce of the probability of failure make to expanding the design speed and stability margin. Uncertainties of material properties and ply thickness for robust optimization design of wings [16] have been used the polynomial chaos expansion (PCE) and gaussian processes (GP) method. This procedure is used to estimate the mean, variance and the probability density function (PDF) of uncertainties and flutter speed in optimal deterministic design for critical speed in flutter/divergence. It successfully used to obtain robust designs rather than the method in group of non-probabilistic, which has only few successive in solving such the problem. From the literature review, in this paper, the PSIBDO with fuzzy uncertainties is used to design a composite aircraft wing structure. The technique is used to quantify epistemic uncertainty due to the lack of knowledge in material properties and allowable transverse displacement. The present study expects to fill the gap left when utilizing the non-probability method for aircraft wing design. In the aeroelastic design, the maximum transverse displacement on the wing, wing lift effectiveness and flutter speed are considered as the design criterion. The optimal aircraft wing design of the composite wing considers an effect of aeroelastic response to design variables of aircraft wing structure that are thickness and composite ply orientations. It leads our research

aiming to study aircraft wing design based on the PSIBDO, which expects to make it more realizable in practical.

The remaining of this paper is organized as follows: the details of RBDO using PSIBDO approach are presented in Section 2. The teaching-learning based optimization (TLBO) is described in Section 3. The design demonstration is performed in Section 4. The design results and conclusions are detailed in Sections 5 and 6, respectively.

## 2. Possibility Safety Index Based Design Optimization (PSIBDO)

### 2.1. Possibility Safety Index Value

A reliability-based design optimization of aircraft wing with design variables (*x*) includes thickness and composite ply orientations. Material property and allowable transverse displacement are uncertainty variable (*a*). In case of RBDO of aircraft wing structure, design constraint $g(x,a) \geq 0$. The aircraft wings will fail when $g(x,a) \leq 0$. The possibility safety index (PSI) value can be presented as:

$$\pi_f = Pos\{g(x, a) \leq 0\} \tag{1}$$

where $Pos\{\ \}$ possibility of an incidence.

To evaluate PSI value, a double-loop nested strategy is required for finding $\pi_f$ or $\alpha$ from Equation (2).

$$Pos(g_i(x, a) \leq 0) = \left\{ \begin{array}{ll} 0, & g_i^{-0} \geq 0 \\ \alpha, & \text{where} \quad g_i^{-0} < 0 < g_i^{-1} \\ 1, & g_i^{-1} < 0 \end{array} \right. \tag{2}$$

where $g_i^{-0}$ and $g_i^{-1}$ are the lower and upper bounds of $g_i(x,a)$ of the membership function in Figure 1, respectively.

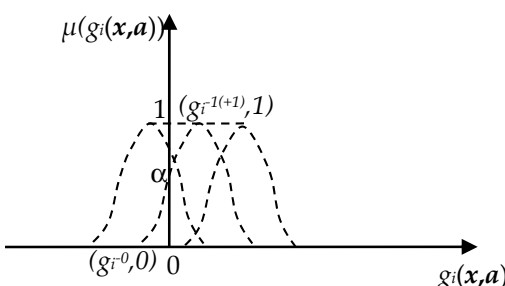

**Figure 1.** Normal membership function $\mu(g_i(x,a))$.

Equation (2) can be solved, if $g_i^{-0}$ and $g_i^{-1}$ are known. If $g_i^{-0} \leq 0$, we can achieve that $Pos(g_i(x, a) \leq 0) = 0$ or 1, and the solution procedure can be finished. In cases that $g_i^{-0} < 0 < g_i^{-1}$, the equation $g_i^{-\alpha} = 0$ should be solved, and its solution $\alpha$ will be the value of $Pos(g_i(x, a) \leq 0)$. The bi-section method is used to compute $\alpha$, which is the value of $\pi_f$. It has the procedure as shown in the Algorithm 1.

---

**Algorithm 1.** Bi-section method

---

1: Initialization

$\alpha_1^0 = 0, \alpha_2^0 = 1$ and specify the termination value as $\varepsilon = 1 \times 10^{-8}$;

2: Iteration 1

2.1: Calculate $g_i^{-\alpha_1^0}$ and $g_i^{-\alpha_2^0}$, and if $g_i^{-\alpha_1^0} \geq 0$ or $g_i^{-\alpha_2^0} \leq 0$ holds,

obtain $\pi_f = 0$ *or* 1 and terminate the iterative procedure. otherwise,

2.2: Calculate $g_i^{-(\alpha_1^0 + \alpha_2^0)/2}$ and go to Step 3

3: Iteration $k$ ($k \geq 1$)

3.1: If $g_i^{-\alpha_1^{k-1}} \times g_i^{-(\alpha_1^{k-1}+\alpha_2^{k-1})/2} > 0$ holds, then let $\alpha_1^k = \left(\alpha_1^{k-1} + \alpha_2^{k-1}\right)/2$ and $\alpha_2^k = \alpha_1^{k-1}$;

If $g_i^{-\alpha_1^{k-1}} \times g_i^{-(\alpha_1^{k-1}+\alpha_2^{k-1})/2} > 1$ holds, then let $\alpha_2^k = \left(\alpha_1^{k-1} + \alpha_2^{k-1}\right)/2$ and $\alpha_1^k = \alpha_1^{k-1}$

3.2: Go to Step 4.

4: Ending

4.1: Calculate the absolute value $\left|\alpha_2^{k-1} - \alpha_1^{k-1}\right|$, and if the ending condition $\left|\alpha_2^{k-1} - \alpha_1^{k-1}\right| \leq \varepsilon$ holds,

4.2: Stop the iterative procedure, and estimate $\pi_f$ by $\pi_f = \left(\alpha_1^{k-1} + \alpha_2^{k-1}\right)/2$; otherwise,

4.3 Return to Step 3 and continue the procedure until the termination

condition $\left|\alpha_2^{k-1} - \alpha_1^{k-1}\right| \leq \varepsilon$ is met.

---

The algorithm is used to find direct PSI value, which is the double-loop nested problem.

*2.2. PSIBDO Problem*

In general, deterministic design of aircraft wing using metaheuristic (MH) optimization can be modelled as shown as follows:

$$Min\ f(\boldsymbol{x}) \tag{3}$$

$$Subject\ to\ g_i(\boldsymbol{x}) \geq 0\ i = 1,\ 2,\ \ldots, N$$
$$\boldsymbol{x}^L \leq \boldsymbol{x} \leq \boldsymbol{x}^U$$

where $x$ is the design variables in $N$-dimensional, $f$ is the objective function for optimization to minimized, and $g_i(\ )$ is the $i$th constraint, $x^L$ and $x^U$ are the lower and upper boundary condition of the design variable. The solution of Equation (3) is said to be a deterministic solution, which is often unrealizable in practice due to the presence of uncertainties [17]. In cases of the uncertainties, it is performed in the form of a fuzzy $(a)$ function. To address such a problem, RBDO is an alternative choice to solving the original deterministic problem. The RBDO cooperates with the fuzzy set theory can be performed with the possibility safety index (PSI) of the constraints [9] and it is called PSIBDO. The formulation can be shown as follows.

$$Min\ f(\boldsymbol{x}) \tag{4}$$

$$Subject\ to\ \ \pi_f = Pos\{g_i(\boldsymbol{x}, \boldsymbol{a}) \leq 0\} \leq \pi_f^{max}, i = 1,\ 2,\ \ldots, N$$
$$\boldsymbol{x}^L \leq \boldsymbol{x} \leq \boldsymbol{x}^U$$

Suppose that the fuzzy uncertainties of aircraft wing structure $(\boldsymbol{a})$ are the Young's modulus $(E)$ and allowable transverse displacement $(u)$. Hence, the PSIBDO of the wing structure considering allowable PSI $\pi_f^{max}$ as constraints can be defined as follows:

$$Min\ f(\boldsymbol{x}) \tag{5}$$

$$Subject\ to\ \ \pi_f = Pos\{g_i(\boldsymbol{x}, \boldsymbol{a}) \leq 0\} \leq \pi_f^{max}, i = 1,\ 2,\ \ldots, N$$
$$\boldsymbol{x}^L \leq \boldsymbol{x} \leq \boldsymbol{x}^U$$
$$0 \leq \pi_f^{max} \leq 1$$

The value of allowable PSI $\pi_f^{max} \in [0,\ 1]$

The proposed PSIBDO leads to some advantages as:

(1) The new technique is in the non-probabilistic technique group, which saves time consumption in uncertainty quantification when compared with the traditional method.
(2) The present technique is eased to cooperate with high-performance metaheuristics (MHs).
(3) It is easy to apply to many real-world optimization problems without with the trappings of gradient calculation methods in the based gradient group.

## 3. TLBO Optimization Design Parameters for Wing Structure

In this paper, PSIBDO solves with a teaching-learning based optimization (TLBO) [18]. It has been extensive study in various problems [19,20], which has advantages due to its free from parameter settings. The process on TLBO is split into two stages are teaching and learning phase. For teacher phase provide all students will follow up with teachers, grading is used to choose the best student and modified position for students to change follow their teachers as:

$$x_{new} = x_{old} + \textit{Difference Mean} \tag{6}$$

$$\textit{Difference Mean}_i = r_i \left( M_{i,best} - T_f M_{i,avg} \right) \tag{7}$$

$$T_f = round \left[ 1 + r_i \right] \tag{8}$$

where $r_i$ is a uniform random number, $r_i \in [0,1]$, $T_f$ is a teaching factor equal to1 or 2, $M_{avg}$ is the mean position of all student members, and $M_{best}$ is the best solution for teacher. The solution in Equation (7) is the difference of the mean position and the best solution which means updated follow up to best position. The learning phase, similar to the teacher phase, expects to improve the performance of the poorer student by learning from a better student. The updated position of the poorer student toward a better student, computed as:

$$\text{If } f(x_i) < f(x_j) \ x_{new} = x_{old} + r_i(x_i - x_j) \tag{9}$$

$$\text{Else, If } f(x_i) > f(x_j) \ x_{new} = x_{old} + r_i(x_j - x_i) \tag{10}$$

After the student phase, the selection operator is performed a new generation population. The teacher phase and student phase are performed until a termination criterion or maximum iteration is met. Algorithm 2 presents a procedure of the TLBO for optimization in this study.

For solving the optimization problem in Equation (5) using TLBO, it becomes a triple-loop nested problem, which is computational burden. However, it is interesting to study its performance in solving aeroelastic design for aircraft wing structure. As mentioned in the previous part that there have been only a few studies that applied the non-probabilistic approach for solving such the design problem.

---

**Algorithm 2.** TLBO procedure

---

Input: Objective function (*fun*), maximum number of generations ($N_G$), population size ($N_P$)

Output: $f^{best}$, $x^{best}$

Initialization:

0: Generate initial student population $N_P\{x^i_{int}\}$ $(i = 1, \ldots, N_P)$ and function evaluations.

**Main steps**

1: For1:$N_G$

{Teacher Phase}

2: Set $x^i_{int}$ from Step 0 (for the 1st loop) or $x^i_l$ from Step 15 as $x^i_{old}$.

3: Compute the mean position of all members, $M_{avg}$.

4: Define the best solution for teacher, $M_{best}$

5: For $i = 1:N_P$

6: Update position $x^i_{new}$ based on Equations (6)–(8).

7: Compute the objective function value of $x^i_{new}$

8: End

9: Select $N_P$ best solution from $x^i_{old} \cup x^i_{new}$

{Student Phase}

10: Set $x^i_{new}$ from the previous step as $x^i_{old}$

11: For $i = 1:N_P$

12: Update position $x^i_{new}$ based on Equations (9) and (10)

13: Compute the objective function value of $x^i_{new}$

14: End

15: Select $N_P$ best solution $x^i_l$ from $x^i_{old} \cup x^i_{new}$

16. End

---

## 4. Numerical Experiment

Aeroelastic design of an aircraft wing structure by fuzzy uncertainties in material property and transverse displacement on the wing is used for a design demonstration. The PSIBDO with fuzzy uncertainty is used to quantify epistemic uncertainty due to lack of knowledge of material properties and allowable transverse displacement. For value of material consider only the Young's modulus in aluminum only since it has a greater impact on the mass of aircraft wing than laminated carbon fiber materials. In this research, the Goland wing is used as a model for design demonstration in Figure 2 [21].

The general geometry of the Goland wing composed of chord length and semi-span wing as 1.216 m and 6.096 m, respectively and wing thickness of ±0.0508 m. The aircraft wing structure is made of aluminum ribs and spars, while the skins are made from three layers of laminated carbon fiber. The properties of both materials are presented in Table 1.

The optimization problem for minimize mass of an aircraft wing subject to aeroelastic and structural constraints is expressed as:

$$Min f(\boldsymbol{x}) = mass \tag{11}$$

Subject to $-u_{max} + u_{al} \geq 0$

$$-V_{f,al} + V_f \geq 0$$
$$-\eta_{L,al} + \eta_L \geq 0$$
$$\boldsymbol{x}^L \leq \boldsymbol{x} \leq \boldsymbol{x}^U$$

where $\boldsymbol{x}$ is $\boldsymbol{a}$ design variable vectors, $x^L$ and $x^U$ is lower and upper bounds, $u_{max}$ is the maximum transverse displacement on the aircraft wing structure, while $u_{al}$ is allowable transverse displacement. $\eta_L$ and $\eta_{L,al}$ are wing lift effectiveness and allowable lift effectiveness, respectively. $V_f$ is a flutter speed and $V_{f,al}$ is an allowable flutter speed. Aerodynamic properties, aeroelastic and structural constraints are set as $\rho_{air}$ = 1.2 kg/m$^3$, free stream velocity = 40 m/s, $u_{al}$ = 0.1 m, $\eta_{L,al} \geq 0.9$ and $V_{f,al} \geq 180$ m/s. The Goland wing opens angle of attach for 3°.

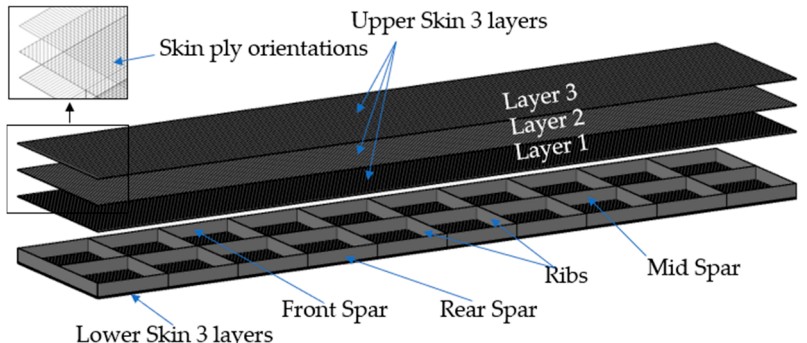

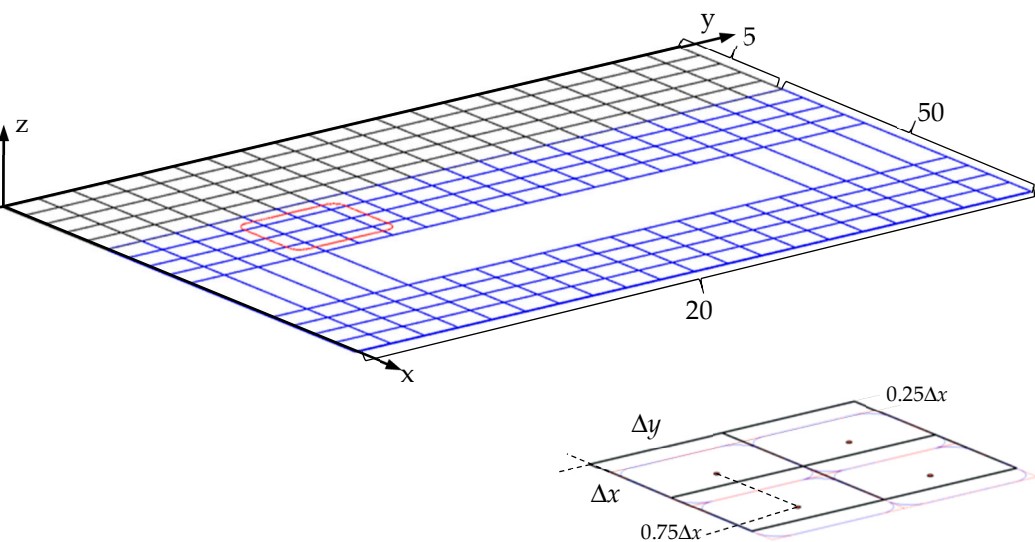

**Figure 2.** Geometry model of a Goland wing and aerodynamic grid.

**Table 1.** Material properties of aluminum and carbon fiber.

| Aluminum | | |
|---|---|---|
| **Properties** | **Value** | **Unit** |
| Young's modulus ($E$) | $70 \times 10^9$ | Pa |
| Poisson's ratio ($\nu$) | 0.3 | - |
| Density ($\rho$) | 2700 | kg/m$^3$ |
| **Carbon fiber** | | |
| **Properties** | **Value** | **Unit** |
| Young's modulus ($E_{11}$) | $1.49 \times 10^{11}$ | Pa |
| Young's modulus ($E_{22}$) | $8.83 \times 10^9$ | Pa |
| Shear modulus ($G_{12}$) | $5.38 \times 10^9$ | Pa |
| Shear modulus ($G_{13}$) | $5.38 \times 10^9$ | Pa |
| Shear modulus ($G_{23}$) | $2.98 \times 10^9$ | Pa |
| Longitudinal tensile strength ($Sty_{11}$) | $500 \times 10^6$ | Pa |
| Transverse tensile strength ($Sty_{22}$) | $5 \times 10^6$ | Pa |
| In-plane tensile strength ($Sty_{12}$) | $35 \times 10^6$ | Pa |
| Longitudinal compressive strength ($Scy_{11}$) | $350 \times 10^6$ | Pa |
| Transverse compressive strength ($Scy_{22}$) | $75 \times 10^6$ | Pa |
| In-plane compressive strength ($Scy_{12}$) | $35 \times 10^6$ | Pa |
| Poisson's ratio ($\nu_{12}$) | 0.342 | - |
| Density ($\rho$) | 1800 | kg/m$^3$ |

Aeroelastic phenomena in this design analysis of the composite aircraft wing can be categorized in two groups, which are static and dynamic aeroelastics. The most significant static aeroelastic parameters are lift effectiveness and divergence speed, while dynamic aeroelastic phenomenon is flutter speed. The divergence is less important when compared with the flutter speed, so it not included in the optimization design problem. Lift effectiveness is the ratio of total wing lift when considering flexibility to the rigid wing counterpart. The lift effectiveness reflects an ability of a wing to produce lift force when a structure is flexible rather than rigid or high-strength. A value above 0.9 is desirable.

The design variables of aircraft wing structure can be divided into two groups that are thickness and composite ply orientations. The total 14 design variables are detailed here:

$x_{1-3}$ = upper skin thickness
$x_{4-6}$ = lower skin thickness
$x_7$ = rib thickness
$x_8$ = spar thickness
$x_{9-11}$ = lower skin ply orientations
$x_{12-14}$ = upper skin ply orientations

To make our model realizable in practice, the design variable vectors are defined to discrete value where the extent constraints as follows:
for $i$ = 1–8,

$x_i \in$ {0.0005 0.001 0.0015 0.002 0.0025 0.003 0.0035 0.004 0.0045 0.005} m
and for $i$ = 9–14,

$x_i \in$ {0 15 30 45 60 75 90 105 120 135 150 165} degree.

The deterministic design optimization problem of the aircraft wing in Equation (11) can be written in the form of a RBDO problem as in Equation (5). In this study, uncertainties (Young's modulus ($E$) and ($u$)) are quantified by a fuzzy set technique. The numbering of constraints $g_i(x, a)$ that is 1st, 2nd and 3rd, are assigned for allowable transverse displacement, allowable flutter speed, and allowable lift effectiveness—respectively. The RBDO problem is presented as follows:

$$Min f\ (\boldsymbol{x}) = mass \tag{12}$$

$$\text{Subject to Pos}\{g_i(\boldsymbol{x}, \boldsymbol{a}) \leq 0\} \leq \pi_{fi}^{max}\ (i = 1,\ 2,\ 3)$$

where $g_1(\boldsymbol{x}, \boldsymbol{a}) = -u_{max} + u_{al}$

$$g_2(\boldsymbol{x}, \boldsymbol{a}) = -V_{f,al} + V_f$$
$$g_3(\boldsymbol{x}, \boldsymbol{a}) = -\eta_{L,al} + \eta_L$$
$$\boldsymbol{x}^L \leq \boldsymbol{x} \leq \boldsymbol{x}^U$$

Aircraft wing structural analysis for both static and dynamics are handled by finite element analysis (FEA), which uses quadrilateral Mindlin shell elements. Aerodynamic analysis is solved using the unsteady vortex lattice method (UVLM) [22], which is known to perform quickly with moderate competency when compared with computational fluid dynamics (CFD), and various forms of potential flow analysis. Aeroelastic analysis needs a surface spline interpolation technique to interface between structural and aerodynamic forces. Reduced order modeling UVLM in the form of a discrete-time aeroelastic model of an aircraft wing has been proven efficient in the analysis of flutter speed for low speed aerodynamics [22]. In reliability-based design optimization as mentioned causes computation burden due to the complexity in analysis of models and uncertainty quantification. The complexity can be alleviated by improving uncertainty quantification using the proposed method in of the non-probabilistic group and reduce the complexity in structural and aeroelasticity analysis. A quasi-steady aerodynamic approach is used for flutter analysis, which gives satisfactory computational results compared to commercial software [14]. The aerodynamic grid on main body wing and wake is presented in Figure 2. Each element panel has chorwise length $\Delta x$ and spanwise width $\Delta y$. The element panels and the relative placements of vortex ring elements are shown as the enlarged view. The main body wing

consists of 5 chordwise and 20 spanwise panels, while the wake sheds from the tailing edge for 50 chordwise panels. The structural mass is an objective function as fuel saving is needed for modern aircraft. It is even more advantageous since it results for the moment of inertia in directional/lateral activation to reducing. Flutter speed is assigned as a constraint function to handle aircraft structural performance. The lift effectiveness is prescriptive that the wing still has available aerodynamic effective appraisal when consideration of flexibility. The last constraints are added to assure for necessary safety in operating conditions.

An optimizer for solving the PSIBDO in this study is TLBO, which is efficient for solving single-objective. It has been proven that the efficiency of the TLBO outperforms other algorithms [20]. The performance caused by the teaching phase is used for exploitation and the learning phase emphasizes exploration. This technique tends to be more efficient with an exploration-based reproduction operator. As a result, the TLBO has been proven in terms of computational performance and convergence. This process may occur in the real-world classes that prepare students to learn from teachers, self-study, or participate with classmates.

## 5. Design Results and Discussions

The uncertainties of material property and allowable transverse displacement on the wing are Young's modulus ($E$) and $u$, respectively. The uncertainties are modelled with normal membership functions in such a way that $E = \exp\left\{ -1/2 \left( \frac{r - Em}{Em \times 0.1} \right)^2 \right\}$ and $u = \exp\left\{ -1/2 \left( \frac{r - Ualm}{Ualm \times 0.1} \right)^2 \right\}$, where $E_m$ = 70 GPa, and $U_{alm}$ = 0.1 m are the core of the functions, respectively. The figure of the normal membership function of $E$ and $u$ are presented in Figure 3. By using the PSIBDO technique, Equation (9) can be changed to Equation (10) using a triple-loop nested to solve the problem using a computer with the following specification: AMD Ryzen 7 3700X with Radeon Graphics 3.00 GHz, 32.00 GB, 64-bit Window 10 operating system. The optimizer performs an initial setting of the algorithm with a population size of 50 and termination iteration of 200.

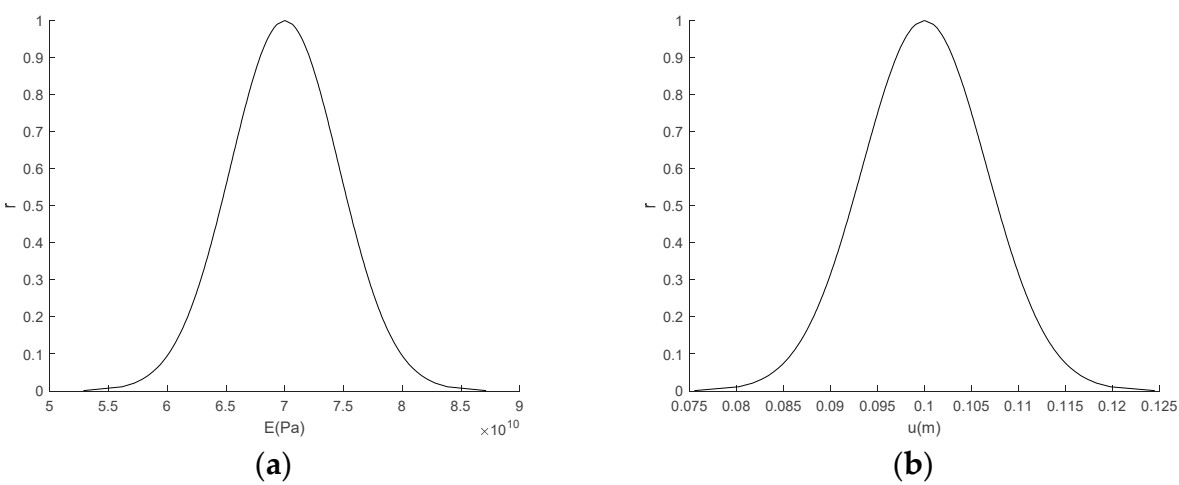

**Figure 3.** Normal membership functions of (**a**) Young's modulus ($E$) and (**b**) allowable transverse displacement ($u$).

Using the suggested PSIBDO, the results of the optimum solutions for the aircraft wing design are given in Table 2. The deterministic results for an optimum design variables are prescribed as PSI $\pi_{fi}{}^{max}$ = 1. The aeroelastic results corresponds to PSIs as less allowable PSIs cause safer aeroelastic characteristics in both static and dynamic as shown in the last column of Table 2.

**Table 2.** Optimum solutions for the aircraft wing design.

| Case | Mass (kg) | Flutter Speed (m/s) | Lift Effectiveness ($\eta$) |
|---|---|---|---|
| $\pi_{fi}{}^{max} = 0.001$ | 103.0109 | 302.5471 | 1.0272 |
| $\pi_{fi}{}^{max} = 0.01$ | 98.1614 | 278.3650 | 1.0578 |
| $\pi_{fi}{}^{max} = 0.1$ | 83.1111 | 251.2743 | 1.1149 |
| Deterministic | 72.4086 | 190.189 | 1.151 |

As exposed by Table 2, the total mass of the aircraft wing structure has an increasing trend with the allowable PSI $\pi_{fi}{}^{max}$ being reduced from the value of 0.1 to 0.001. Meanwhile, the critical speed increases with the allowable PSI $\pi_{fi}{}^{max}$ being reduced, and the lift effectiveness decreases with the allowable PSI $\pi_{fi}{}^{max}$ reducing. Figure 4 demonstrates the comparison of search history for $\pi_{fi}{}^{max}$ = 0.1, 0.01 and 0.001, respectivelyand presents the upper and lower skin ply orientations. When running TLBO for solving the problem, the iterative procedure stops when the number of iterations reaches 100. It equals termination iteration 200 due to the nature of TLBO, which has embedded teaching and learning phases. The results in Table 3 show the total mass of the aircraft wing structure have an increasing trend with the allowable PSI $\pi_{fi}{}^{max}$ reducing. Moreover, the skin thicknesses in Layers 2 and 5 increase with the allowable PSI $\pi_{fi}{}^{max}$ reducing, while the skin thicknesses in Layers 1, 3, 4, and 6—rib thicknesses and spar thicknesses approach to their lower limits. The values of the skin ply orientations between layer increases with the allowable PSI $\pi_{fi}{}^{max}$ reducing. Furthermore, the results in Table 3 and the search history in Figure 4 reveal that the search history in Figure 4 are already constant before iteration 40 and that the remaining iterations do not yield significant benefits. The chosen ply thickness with increasing steps of 0.5 mm to 5 mm is not enough to bring the mass reduction as shown. The optimum thickness varies by two different thicknesses (0.5 and 1.0 mm). For future study, the step change should be reduced to 0.1 mm to provide an optimized search for a more optimum mass.

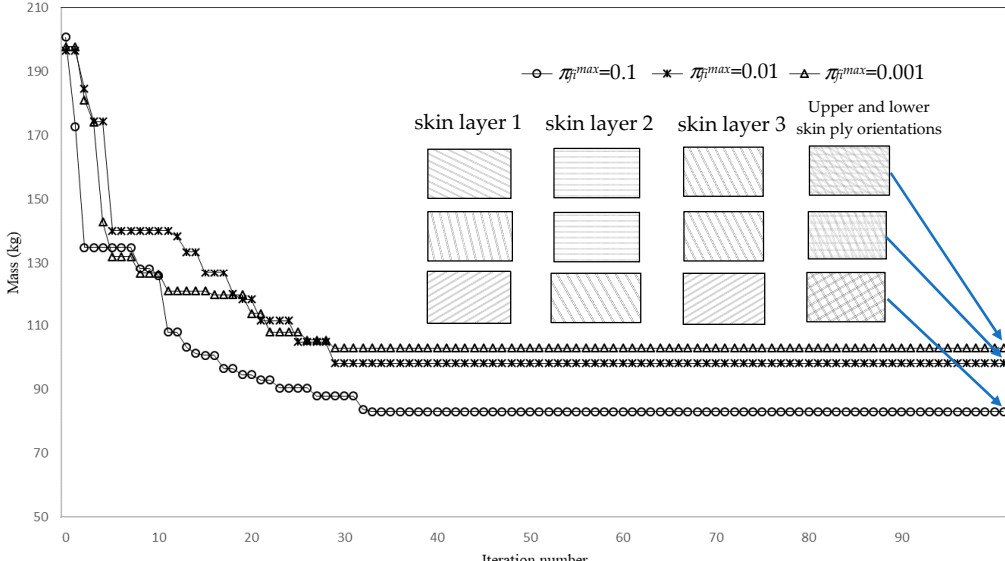

**Figure 4.** The search history of the composite aircraft wing design.

The optimum thickness varies in range from 0.5–1.0 mm, which eliminates the practical thickness. It should be more than 3 mm to maintain resistance damage [23], otherwise, the ply orientation might still change. For upper and lower skins using a laminated carbon fiber with three layers, from the calculated results, a symmetric and equivalent laminate for upper and lower skins in each PSI $\pi_{fi}{}^{max}$ = 0.1, 0.01 and 0.001 with the orientations of [30°/120°/30°], [105°/0°/120°] and [150°/0°/120°], respectively as displayed in Figure 5. The optimum ply orientation can eliminate shearing and bending twist coupling behaviors.

**Table 3.** Design variables of optimum solutions for the aircraft wing design by PSIBDO.

| Design Variables and Objective Function | $\pi_{fi}{}^{max} = 0.1$ | $\pi_{fi}{}^{max} = 0.01$ | $\pi_{fi}{}^{max} = 0.001$ | Deterministic |
|---|---|---|---|---|
| Upper skin Layer 1 (mm.) | 0.0005 | 0.0005 | 0.0005 | 0.0005 |
| Upper skin Layer 2 (mm.) | 0.0005 | 0.0005 | 0.001 | 0.0005 |
| Upper skin Layer 3 (mm.) | 0.001 | 0.001 | 0.001 | 0.0005 |
| Lower skin Layer 1 (mm.) | 0.0005 | 0.0005 | 0.0005 | 0.0005 |
| Lower skin Layer 2 (mm.) | 0.0005 | 0.001 | 0.001 | 0.0005 |
| Lower skin Layer 3 (mm.) | 0.001 | 0.001 | 0.001 | 0.0005 |
| Rib thickness (mm.) | 0.0015 | 0.0015 | 0.0015 | 0.001 |
| Spar thickness (mm.) | 0.001 | 0.001 | 0.001 | 0.001 |
| Lower skin Layer 1 (deg.) | 30 | 105 | 150 | 135 |
| Lower skin Layer 2 (deg.) | 120 | 0 | 0 | 60 |
| Lower skin Layer 3 (deg.) | 30 | 120 | 120 | 135 |
| Upper skin Layer 1 (deg.) | 30 | 105 | 150 | 135 |
| Upper skin Layer 2 (deg.) | 120 | 0 | 0 | 60 |
| Upper skin Layer 3 (deg.) | 30 | 120 | 120 | 135 |
| Mass (kg) | 83.1111 | 98.1614 | 103.0109 | 72.4086 |

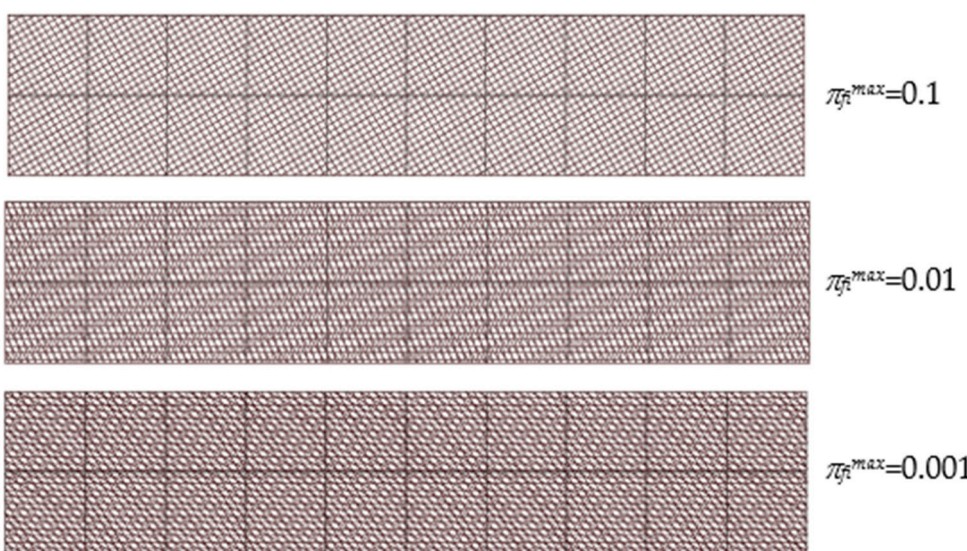

**Figure 5.** Layout of ply orientation angles on upper and lower wing skin in the composite laminate.

The first 10 lowest natural frequency modes used in flutter analysis at each PSI value are presented in the Table 4. From the results shown, the natural frequencies are very high for a wing—even for wind tunnel model testing. This is caused by the strict deformation constraint of 0.1 m, which is less than 2% of half the wings pan. The plots of damping ratio versus velocity with various PSI $\pi_{fi}{}^{max}$ values are shown in Figure 6. The results show flutter speeds estimated by quasi-steady aerodynamics, which are affected by PSI $\pi_{fi}{}^{max}$ as the lower PSI $\pi_{fi}{}^{max}$ results in the lower flutter speeds.

Furthermore, from comparing the design variables and the objective functions, the result from PSIBDO is larger than the deterministic solution. The results may indicate that skin thicknesses in Layers 1, 3, 4, and 6, and spar thickness to lower stress when the aircraft wing is subject to the aerodynamic loads.

**Table 4.** The PSI $\pi_{fi}{}^{max}$ various on natural frequency of the wing (Hz).

| Mode | PSI $\pi_{fi}{}^{max}$ = 0.1 | PSI $\pi_{fi}{}^{max}$ = 0.01 | PSI $\pi_{fi}{}^{max}$ = 0.001 |
|---|---|---|---|
| 1 | 99 | 100 | 138 |
| 2 | 1605 | 1845 | 2400 |
| 3 | 2844 | 2925 | 3810 |
| 4 | 7096 | 7550 | 7758 |
| 5 | 7236 | 12,310 | 17,538 |
| 6 | 16,402 | 26,720 | 31,310 |
| 7 | 25,796 | 29,036 | 32,960 |
| 8 | 30,601 | 33,360 | 61,000 |
| 9 | 51,157 | 57,687 | 86,510 |
| 10 | 77,410 | 86,321 | 102,920 |

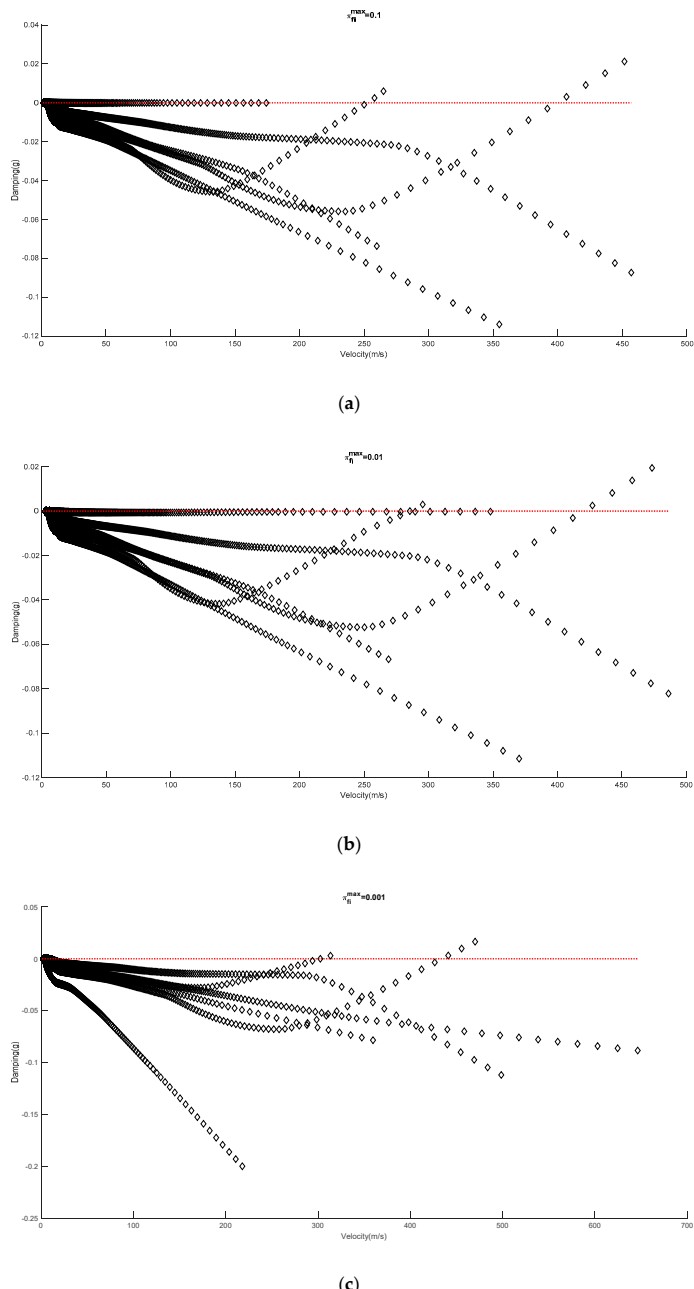

**Figure 6.** The variations of the PSI at (a) $\pi_{fi}{}^{max}$ = 0.1, (b) $\pi_{fi}{}^{max}$ = 0.01 and (c) $\pi_{fi}{}^{max}$ = 0.001 due to the variation damping ratio vs. velocity.

## 6. Conclusions

Reliability-based design optimization of an aircraft wing uses a fuzzy-based MH approach. It is known that the optimum aircraft structure may be unrealizable due to uncertainties. This is means uncertainties should be taken into account in our design problem. The main objective of this work is to demonstrate a non-probabilistic approach to design of aero structures. The present study uses a fuzzy technique to model the uncertainties and find allowable PSI of the problem. The problem is the triple-loop nested problem, which causes computational burden. The performance of this technique in design of aircraft wing aeroelasticity is needed to quantify. From the numerical results, it shows the PSIBDO can generate conservative optimal aircraft structures depending on the PSI values. Results which are too conservative can be confined by the expert opinion.

For the future study, adaptation of PSIBDO approach is needed to study for reducing complexity in analysis by the way of multi-objective optimization, which is real world problem rather than the single one. It can find a solution set in one optimization run. The various aero-structures of aircraft are our purpose in design, which we expect to make more practical.

**Author Contributions:** Conceptualization, S.W., S.S. and S.B.; methodology, S.W. and S.S.; setup and designed the numerical experiments, S.W., S.S. and S.B.; performed the numerical experiments; S.W. and S.S.; analyzed the data S.W.; writing—original draft, S.W.; writing—review and editing, S.W., S.S. and S.B. All authors have read and agreed to the published version of the manuscript.

**Funding:** The authors are grateful for the financial support provided by King Mongkut's Institute of Technology Ladkrabang, and the National Research Council Thailand.

**Institutional Review Board Statement:** Not applicable.

**Informed Consent Statement:** Not applicable.

**Data Availability Statement:** The authors confirm that the data supporting the findings of this study are available within the article.

**Acknowledgments:** The authors are grateful for the financial support provided by King Mongkut's Institute of Technology Ladkrabang; the National Research Council Thailand; and the Doctoral Program from Research, Graduate School, KhonKaen University.

**Conflicts of Interest:** The authors declare no conflict of interest. The funders had no role in the design of the study; in the collection, analyses, or interpretation of data; in the writing of the manuscript, or in the decision to publish the results.

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
