# Peer review of "Reliability-Based Design of an Aircraft Wing Using a Fuzzy-Based Metaheuristic"

_applsci, doi:10.3390/app11146463_

Round 1

Reviewer 1 Report

The paper provides a good overview over a discrete optimization problem of a composite wing with uncertainties. The result section gives a good insight on how the various possibility safety indexes influence the optimized designs. Nevertheless, I would like to address several points as written below.

General comments:
- Please revise grammar and spelling, as well as missing/double space characters.
- I understand that unidirectional composite plies have certain minimum thicknesses that have to be complied with. With the chosen ply thickness of 0.5 mm (page 6) however, the optimization results listed in Table 3 show only two different values for the skin thicknesses (0.5 and 1.0 mm). Moreover, the mass results of Figure 2 are already constant before iteration 40, so that the remaining iterations do not bring mass benefits anymore (while the ply angles might still change). My question is: would it be more sensible to assume a smaller thickness increment (e.g. 0.1 mm), so that the optimization results could provide a deeper insight, show more variations (more different thicknesses) and smaller steps in the changes of mass? I don't think that a new optimization run is necessary, but could you please comment on that in the manuscript?

Specific comments:
- Page 3 below Eq. 1: I could not find the term sup{.} in any equation, is the definition of sup{.} necessary?
- Eq. 2: Please provide the definition of alpha and the superscript indexes of g_i (-0 and -1).
- Eq. 8: If r_i is a random number, how reproducable is one certain wing design in a new optimization run, where the random numbers would be different ones?
- Eq. 9 and 10: I think there should be a difference between the if conditions f(x_i)<f(x_j) in both equations?
- Figure 1: How does the aerodynamic grid look compared to the structure? Do the front and rear spars coincide with the leading and trailing edge respectively, or is there some space between the front spar and leading edge, as well as the rear spar and trailing edge?
- Table 1: How come is the density of aluminium set to 4000 kg/m^3 instead of 2800 kg/m^3?
- Eq. 11: What is the definition of wing lift effectiveness; does a value of 1.0 represent the lift slope of a rigid wing?
- Page 6 below Eq. 11: At which angle of attack or lift coefficient does the deformation constraint of 0.1 m have to be fulfilled?
- Page 7 below Eq. 12: "The structural mass is an objective function as energy saving is needed for modern aircraft." I would rather write fuel saving instead of energy saving.
- Section 5: Could you please provide a figure with the distribution functions of the Young's modulus and the allowable displacement? That would help in getting an overview and a sense of numbers quicker.
- Table 4: (Comment) Those natural frequencies are very high for a wing (even for a wind tunnel model). I suppose that this is caused by the strict deformation constraint of 0.1 m (less than 2% of half span).
- Figure 4: I see that between 250 m/s and 300 m/s there are flutter curves that might cross the zero damping line. Could those be the relevant flutter speeds instead of the 400+ m/s?

Author Response

REVIEWER 1

Comments and suggestions for authors

The paper provides a good overview over a discrete optimization problem a composite wing with uncertainties. The result section gives a good insight on how the various possibility safety indexes influence the optimized designs. Nevertheless, I would link to address several points as written below.

General comments:

-please revise grammar and spelling, as well as missing/double space characters.

Reply:We have improved grammar and spelling by an English native speaker. Thank you for your suggestion.

- I understand that unidirectional composite plies have certain minimum thicknesses that have to be complied with. With the chosen ply thickness of 0.5 mm. (page 6) however, the optimization results listed in table 3 show only two different values for the skin thicknesses (0.5 and 1.0 mm.). moreover, the mass results of figure 2. Are already constant before iteration 40, so that the remaining iterations do not bring mass benefits anymore (while theply angles might still change). My question is: would it be more sensible to assume a smaller thickness increment (e.g. 0.1 mm.), so that the optimization results could provide a deeper insight, show more variations (more different thicknesses) and smaller steps in the changes of mass. I don’t think a new optimization run is necessary, but could you please comment on that in the manuscript.

Reply:We have added the comments into the section of design results and discussions. Thank you for your suggestion.

Specific comments:

-page 3 below Eq.1: I could not find the term sup{.} in any equation, is the definition of sup{.} necessary.

Reply:We have addressed the mistake.

Thank you for your suggestion.

- Eq. 2: please provide the definition of alpha and the superscript indexes of g_i (-0 and -1).

Reply: we have added the definition of alpha and and .

Thank you for your suggestion.

- Eq. 8: if r_i is a random number, how reproducible is one certain wing design in a new optimization run, where the random numbers would be different ones.

Reply:TLBO is the one of the best known metaheuristics. Such an optimizer relies on randomization and is classified as soft computing. With many optimization runs, the results are unlikely to be exactly the same but they are similar to some extent. The more consistent metaheuristic means they can reproduce somewhat similar results. In order to cope with such a drawback, we need to run TLBO with a high number of iterations. In this work, we use 200 iterations for one run, which is sufficient as can be seen in the search history in Figure 4.

-Eq. 9 and 10: I think there should be a difference between the if conditions f(x_i)<f(x_j) in both equations.

Reply: we have addressedthe mistake. Thank you for your suggestion.

-figure 1: how does the aerodynamic grid look compared to the structure. Do the front and rear spars coincide with the leading and trailing edge respectively, or is there some space between the front spar and leading edge, as well as the rear spar and trailing edge.

Reply: we have added the aerodynamic grid and the explanation to make it clearer.

Thank you for your suggestion.

-table 1: how come is the density of aluminium set to 4000 instead of 2800

Reply:We sincerely apologize for the mistake in typing. We have corrected the density value.

Thank you for your suggestion.

Eq. 11: what is the definition of wing lift effectiveness; does a value of 1.0 represent the lift slope of a rigid wing.

Reply: We have added the definition of the wing lift effectiveness at below of eq.11.

Thank you for your suggestion.

-page 6 below Eq. 11: at which angle of attack or lift coefficient does the deformation constraint of 0.1 m. have to be fulfilled.

Reply: We have added angle of attack in our analysis into the numerical experiment section.

Thank you for your suggestion.

-page 7 below Eq. 12: “the structural mass is an objective function as energy saving is needed for modern aircraft”. I would rather write fuel saving instead of energy saving.

Reply: We have corrected the word.

Thank you for your suggestion.

-section 5: could you please provide a figure with the distribution function of the Young’s modulus and the allowable displacement. That would help in getting an overview and a sense of to be numbers quicker.

Reply:We have added the normal membership function of Young’s modulus, E and the allowable displacement, u to make it clearer.

Thank you for your suggestion. 

-table 4: (comment) those natural frequencies are very high for a wing (even for a wind tunnel model). I suppose that this is caused by the strict deformation constraint of 0.1 m. (less than 2% of half span).

Reply:We have added the suggestion into the part of experimental results and discussions.

Thank you for your suggestion.

-figure 4: I see that between 250 m/s and 300 m/s there are flutter curves that might cross the zero damping line. Could those be the relevant flutter speeds instead of the 400+ m/s.

Reply: We have updated the flutter speeds in Table 2. The exact flutter speeds are more conservative than the previous, while the trend does not change.

Thank you for your suggestions.

Reviewer 2 Report

My comments are the following:

  1. It is not stated clearly in the in the introduction what uncertainties are exactly treated and that is the scope of these uncertainties.
  2. The motivation of the PSIBDO is not clearly given, what are the main benefits/drawbacks compared to the other existing techniques in the literature?
  3. Is there guarantee that the proposed TLBO procedure converges?
  4. Very little is given about the aerodynamics analysis. Why was the unsteady vortex ring method selected?
  5. How is the fluid/structure interaction done?

I recommend resubmitting the article after addressing the comments above.

Author Response

REVIEWER 2

Comments and suggestions for authors

My comments are the following:

  1. It is not stated clearly in the introduction what uncertainties are exactly treated and that is the scope of these uncertainties.

Reply: we have added the definition of uncertainty and mention the uncertainty that this paper is focusing to treat in the introduction and numerical experiment sections.

Thank you for your suggestion. 

  1. The motivation of the PSIBDO is not clearly given, what are the main benefits/drawbacks compared to the other existing techniques in the literature.

Reply: We have added the advantages of the PSIBDO into the second part (PSIBDO).

Thank you for your suggestion.

  1. Is there guarantee that the proposed TLBO procedure converges.

Reply: We have added some explanation of its initial setting and termination criterion of the TLBO algorithm and its convergence efficiency.

  1. Very little is given about the aerodynamics analysis. Why was the unsteady vortex ring method selected?

Reply: We have added the reason to select the unsteady vortex ring method in the present research.

  1. How is the fluid/structure interaction done?

Reply: We have added the explanation of the fluid/structure interaction in numerical experiment section.

Round 2

Reviewer 2 Report

The authors gave detailed explanation to all of my questions from the first version of the paper. All these details are in the current draft of the paper, therefore i recommend accepting the paper as is.